# Lignans from the Twigs of *Litsea cubeba* and Their Bioactivities

**DOI:** 10.3390/molecules24020306

**Published:** 2019-01-16

**Authors:** Xiuting Li, Huan Xia, Lingyan Wang, Guiyang Xia, Yuhong Qu, Xiaoya Shang, Sheng Lin

**Affiliations:** 1Beijing Advanced Innovation Center for Food Nutrition and Human Health, Beijing Technology and Business University, Beijing 100048, China; lixt@btbu.edu.cn; 2State Key Laboratory of Bioactive Substance and Function of Natural Medicines, Institute of Materia Medica, Chinese Academy of Medical Sciences and Peking Union Medical College, Beijing 100050, China; xiahuan@imm.ac.cn (H.X.); wanglingyan@imm.ac.cn (L.W.); xiaguiyang@imm.ac.cn (G.X.); qyhcxl28@126.com (Y.Q.); 3Beijing Key Laboratory of Bioactive Substances and Functional Foods, Beijing Union University, Beijing 100023, China

**Keywords:** *Litsea cubeba*, cytotoxicity, isolation and elucidation, lignans

## Abstract

*Litsea cubeba*, an important medicinal plant, is widely used as a traditional Chinese medicine and spice. Using cytotoxicity-guided fractionation, nine new lignans **1**–**9** and ten known analogues **10**–**19** were obtained from the EtOH extract of the twigs of *L. cubeba*. Their structures were assigned by extensive 1D- and 2D-NMR experiments, and the absolute configurations were resolved by specific rotation and a combination of experimental and theoretically calculated electronic circular dichroism (ECD) spectra. In the cytotoxicity assay, 7′,9-epoxylignans with feruloyl or cinnamoyl groups (compounds **7**–**9**, **13** and **14**) were selectively cytotoxic against NCI-H1650 cell line, while the dibenzylbutyrolactone lignans **17**–**19** exerted cytotoxicities against HCT-116 and A2780 cell lines. The results highlighted the structure-activity relationship importance of a feruloyl or a cinnamoyl moiety at C-9′ or/and C-7 ketone in 7′,9-epoxylignans. Furthermore, compound **11** was moderate active toward protein tyrosine phosphatase 1B (PTP1B) with an IC_50_ value of 13.5 μM, and compounds **4**–**6**, **11** and **12** displayed inhibitory activity against LPS-induced NO production in RAW264.7 macrophages, with IC_50_ values of 46.8, 50.1, 58.6, 47.5, and 66.5 μM, respectively.

## 1. Introduction

Plants from the *Litsea* species (Lauraceae) are widely distributed in tropical or subtropical areas. *Litsea cubeba*, mainly grown in the east and south of China, is broadly used as a traditional Chinese medicine and spice. “Bi-cheng-qie” and “dou-chi-jiang”, the dried fruits and roots of *L. cubeba*, respectively, have been documented in the Chinese Pharmacopoeia and *Chinese Materia Medica* as two important traditional Chinese medicines for the treatment of various ailments, including coronary disease, cerebral apoplexy, asthma, and rheumatic arthritis [1,2,3]. Moreover, *Litsea cubeba* fruits are also important spices and great sources of essential oils which are often used as flavor enhancers in foods, cigarettes, and cosmetics [4]. Previous phytochemical investigation of the fruits and roots of *L. cubeba* have reported the discovery of aporphine-type alkaloids, lignans, and phenolic constituents [5,6,7,8,9,10,11]. Among them, aporphine-type alkaloids and lignans were considered as the major active principles of this plant due to their antithrombotic, anti-inflammatory, and antinociceptive properties [8,9,12,13,14,15]. Since there are few reports on the phytochemicals of twigs of *L. cubeba*, a recent study on *L. cubeba* twigs by our group led to the characterization of 36 aromatic glycosides from the the water-soluble fraction of an ethanolic extract. Interestingly, some lignan glycosides showed potent hepatoprotective and HDAC1 inhibitory activity [16,17]. In the present study, we have investigated the constituents of the EtOAc-soluble fraction of the ethanolic extract of *L. cubeba* twigs. Bioassay-guided isolation of a fraction with cytotoxicity against HCT-116, NCI-H1650, and A2780 cell lines (IC_50_ = 28.3, 11.5, and 16.8 µg/mL, respectively) led to the discovery of nine new lignans **1**–**9** and ten analogues **10**–**19** (Figure 1). The structures of **1**–**9** were elucidated by spectroscopic methods, and their absolute configurations were determined by optical rotations and a combination of experimental and theoretically calculated electronic circular dichroism (ECD) spectra. Detailed herein are the isolation, structural elucidation, and bioactivity assay of compounds **1**–**19**.

## 2. Results and Discussions

### 2.1. Structure Elucidation

The EtOAc extract of the twigs of *L. cubeba* was subjected to column chromatography on silica gel to give 13 fractions (F_1_–F_13_). Cytotoxicity assays found that F_9_ displayed potent activities against HCT-116, NCI-H1650, and A270 cell lines. Fractionation of F_9_ by Sephadex LH-20, RP-18, preparative TLC, and preparative HPLC led to the discovery of nine new lignans **1**–**9** and the ten known ones **10**–**19**.

Compound **1** was obtained as a white amorphous powder. The presence of amide (1643 cm^−1^), aromatic ring (1611, 1516, and 1459 cm^−1^), and hydroxy (3372 cm^−1^) functionalities were evident in its IR spectrum. Its molecular formula of C_30_H_33_NO_9_ with fifteen degrees of unsaturation was established by HREIMS based on the [M + H]^+^ ion at *m/z* 552.2234 (calcd. 552.2228) and ^13^C-NMR spectrum. In the ^1^H-NMR spectrum recorded in acetone-*d*_6_, the signals for an aromatic singlet integrated for two protons at δ 6.39 (2H, s, H-2′ and H-6′), a methoxy singlet integrated for six protons at δ 3.67 (6H, s, OMe×2), suggested a 1-substituted-3,5-dimethoxy-4-hydroxybenzene ring in **1**. Signals of a singlet proton at δ 6.74 and two methoxy protons at δ 3.86 and 3.58 revealed a pentasubstituted aromatic ring attached two methoxy groups. These ^1^H-NMR signals, together with another two singlet protons at δ 7.19 and 4.62, were indicative of a typical skeleton of 2,7′-cyclolignan-7-en such as thomasic acid [18]. Additionally, the ^1^H-NMR spectrum of **1** displayed characteristic signals for a tyramine group with resonances at δ_H_ 6.98 (2H, d, *J* = 8.5 Hz, H-2” and H-6”), 6.71 (2H, d, *J* = 8.5 Hz, H-3” and H-5”), 2.69 (2H, t, *J* = 7.5 Hz, H_2_-7”), and 3.39 (2H, dt, *J* = 7.5, 4.5 Hz, H_2_-8”). The ^13^C-NMR spectrum of **1** displayed 30 carbon signals, of which twelve could be assigned to be a tyramine moiety (δ_C_ 131.2, 130.5 × 2, 116.0 × 2, 156.6, 35.6, 42.2) and four methoxy groups (δ_C_ 56.6 × 2, 56.5, 60.4), and the remaining eighteen carbons were consistent with the 2,7′-cyclolignan-7-en skeleton. The complete ^1^H- and ^13^C-NMR assignments of **1** were made by a combination of 1D- and 2D-NMR experiments. In the HMBC spectrum of **1**, the two or three bonds long range correlations from H-6 to C-2, C-4, and C-7, from H-7 to C-2, C-6, C-9, and C-8′, from H-7′ to C-3, C-8, C-2′ (C-6′), and C-9′, from H-8′ to C-2, C-7, C-9, and C-1′, from H_2_-9′ to C-8 and C-7′, and from the methoxy protons at δ_H_ 3.58 to C-3′ (C-5′) (Figure 2) confirmed the 2,7′-cyclolignan-7-en type lignan containing a 3,5-dimethoxy-4-hydroxy-benzene moiety. The NOESY correlation observed between H-6 and the methoxy protons at δ_H_ 3.86 together with the HMBC correlation observed for these methoxy protons and C-5 gave the evidence for the location of one methoxy group at C-5. Key HMBC cross-peaks, such as between methoxy protons at δ_H_ 3.58 and C-3, as well as between OH proton at δ_H_ 7.76 and C-4, served to locate this methoxy and OH group at C-3 and C-4, respectively. Furthermore, the tyramine was linked to C-9 to form an amine bond, according to the HMBC correlations from both H_2_-8″ and NH proton to C-9. Therefore, these data completed the planar structure of **1** as *N*-[2-(4-hydroxyphenyl)-ethyl]-4,4′,9′-trihydroxy-3,5,3′,5′-tetramethoxy-2,7′-cyclolignan-7-en-9-amide. H-7′ appearing as a singlet suggested the dihedral angle for the vicinal protons of H-7′ and H-8′ was nearly 90°, requiring a *trans* relationship of H-7′ and H-8′. This assignment was also supported by the NOESY correlations of H-7′ with H_2_-9′, and H-8′ with H-2′ (H-6)′. Finally, the negative optical rotation of **1** demonstrated the 7′*R*,8′*S* absolute configuration of **1** [18,19]. Hence, compound **1** was defined as (−)-(7′*R*,8′*S*)-*N*-[2-(4-hydroxyphenyl)-ethyl]-4,4′,9′-trihydroxy-3,5,3′,5′-tetramethoxy-2,7′-cyclolignan-7-en-9-amide.

Compound **2** was isolated as a white amorphous powder. The IR spectrum exhibited absorptions of hydroxy (3362 cm^−1^), amide (1649 cm^−1^), and aromatic (1612 and 1516 cm^−1^) moieties. Its molecular formula was deduced as C_39_H_42_N_2_O_11_ from the negative HRESIMS at *m/z* 713.2719 [M − H]^−^ (calcd. 713.2716) and the ^13^C-NMR spectrum. This indicated twenty degrees of unsaturation. The NMR spectra of **2** were very similar to those of compound **10**, a known lignan diamide that was also isolated from this plant [20], with the only difference being the replacement of one of a tyramine group by a 3-methoxytyramine moiety (Table 1; Table 2). In the HMBC spectrum of **2**, H_2_-7”′ showed HMBC correlations with the amide carbon at δ_C_ 171.4, which indicated that the 3-methoxytyramine moiety was connected to C-9′ via an amide bond (Figure 2). In the 1D NOE difference spectrum of **2**, H-8′ was enhanced upon irradiation of H-2′ (H-6′). This enhancement, together with H-7′ presented in a singlet, revealed a *trans*-vicinal orientation of H-7′ and H-8′. Finally, on the basis of the negative optical rotation of **2** and biosynthetic considerations, the structure of compound **2** was defined as (−)-(7′*R*,8′*S*)-*N*^1^-[2-(4-hydroxyphenyl)-ethyl]-*N*^2^-[2-(4-hydroxy-3-methoxyphenyl)-ethyl]-4,4′-dihydroxy-3,5,3′,5′-tetramethoxy-2,7′-cyclolignan-7-en-9,9′-diamide. 

Compound **3** gave the same molecular formula, C_39_H_42_N_2_O_11,_ as that of **2** by analysis of the HRESIMS. Compound **3** shared almost identical UV, IR, and ^1^H- and ^13^C-NMR features to those of **2**, which suggested that they both contained the 4,4′-dihydroxy-3,5,3′,5′-tetramethoxy-2,7′-cyclolignan- 7-en-9,9′-diamide core, a tyramine, and a 3-methoxytyramine moieties. 

Further analysis of 2D-NMR data permitted the tyramine and 3-methoxytyramine moieties to be located at C-9′ and C-9 in **3**, the reverse of **2**, via the amide bonds (Figure 2), respectively. Analysis of the 1D NOE difference spectrum of **3** and its optical rotation indicated that **3** had the same absolute configuration as **2**. Therefore, the structure of **3** was confirmed as (−)-(7′*R*,8′*S*)-*N*^1^-[2-(4-hydroxy-3-methoxyphenyl)-ethyl]-*N*^2^-[2-(4-hydroxyphenyl)-ethyl]-4,4′-dihydroxy-3,5,3′,5′-tetramethoxy-2,7′-cyclolignan-7-en-9,9′-diamide.

Compound **4** was obtained as a yellow solid and its molecular formula was deduced as C_31_H_36_O_10_ from HRESIMS. The IR spectrum exhibited absorption bands at 3391, 1608, and 1516 cm^−1^ due to the aromatic and hydroxy groups. The NMR data of **4** showed signals similar with secoisolariciresinol (Table 1; Table 2) [21,22]. However, both the H_2_-9 and C-9 were shifted downfield when compared with secoisolariciresinol. Besides, the ^1^H- and ^13^C-NMR signals attributed to a *trans*-cinnamyloxy unit were present (Table 1; Table 2). These were consistent with the substitution of the *trans*-cinnamyloxy at C-9, which was verified by the key HMBC correlation from H_2_-9 to C-9″. The positive optical rotation of **4** supported the same (8*S*,8′*S*) configuration as that of the known compound (+)-(8*S*,8′*S*)-9,9′-di-*O*-(*E*)-feruloylsecoisolariciresinol (**11**), which has been also isolated from this plant [12]. The (8*S*,8′*S*) configuration was confirmed by the evidence that compound **4** showed optical rotation opposite to that of (−)-1-*O*-feruloylsecoisolariciresinol [21]. Thus, the structure of **4** was defined as (+)-(8*S*,8′*S*)-9-*O*-(*E*)-cinnamoylsecoisolariciresinol. 

The molecular formula of compound **5** was C_32_H_38_O_11_ from the HRESIMS data. Analysis of the 1D- and 2D-NMR data revealed that its planar structure was completely identical to the known lignan, (−)-(8*R*,8′*R*)-9-*O*-(*E*)-feruloyl-5,5′-dimethoxysecoisolariciresinol, but their specific rotation was inverse [23]. Taking into account that **4** was the 5-methoxy analogue of **5** and they displayed similar specific rotation, it is proposed that they both have the (8*S*,8′*S*) configuration. Thus, the structure of **5** was defined as (+)-(8*S*,8′*S*)-9-*O*-(*E*)-feruloyl-5,5′-dimethoxysecoisolariciresinol. 

The planar structure of **6** was proved to be identical to (−)-(8*R*,8′*R*)-9-*O*-(*E*)-feruloyl-secoisolariciresinol (different nomenclature was used in literature [21]) after analysis of the HRMS, and 1D- and 2D-NMR data of **6**. However, the optical rotation of **6** was opposite for (−)-(8*R*,8′*R*)-9-*O*-(*E*)-feruloyl-secoisolariciresinol [21]. Thus, the structure of **6** was determined as (+)-(8*S*,8′*S*)-9-*O*-(*E*)-feruloyl-secoisolariciresinol.

Compound **7**, an amorphous powder, was determined to have the molecular formula of C_32_H_34_O_12_ by HRESIMS. The NMR spectra of **7** were similar to the co-occurring (+)-9′-*O*-*trans*-feruloyl-5,5′-dimethoxylariciresinol (**13**) [24], with the only difference being the replacement of the CH_2_ group by a ketone. These data demonstrated the presence of a ketone moiety at C-7 in **7**. This inference was confirmed by the HMBC cross-peak of H-2(6)/C-7, H_2_-9/C-7, and H-8′/C-7. The coupling constant of H-7′ (*J* = 7.5 Hz) indicated a *trans* relationship of H-7′/H-8′. The presence of correlations of H-7′/H_2_-9′ and H-2(6)/H-8′ and the absence of H-8/H_2_-9′ were observed in the NOESY spectrum of **7**, which confirmed that H-7′ was oriented opposite to H-8 and H-8′. The absolute configuration of **7** was established by quantum chemical ECD calculation (Appendix A). The calculated ECD curve for 8*R*,7′*S*,8′*R*-isomer matched well with the experimental ECD spectrum of **7** (Figure 3), which suggested compound **7** had the (8*R*,7′*S*,8′*R*) absolute configurations. Based on these observations, the structures of **7** was assigned as (+)-(8*R*,7′*S*,8*R*′)-9′-*O*-(*E*)-feruloyl-5,5′-dimethoxylariciresinol-7-one.

The molecular formula of compound **8** was C_33_H_38_O_12_ as indicated by the HRESIMS. The NMR spectra of **8** and (+)-9′-*O*-*trans*-feruloyl-5,5′-dimethoxylariciresinol were closely comparable [24], except for the replacement of (*E*)-feruloyl group by the (*E*)-cinnamoyl group. The structure of **8** was confirmed by the 2D-NMR HSQC, COSY, HMBC, and NOESY data. Also, the NOESY correlations of H-7′/H_2_-9′ and H_2_-7/H_2_-9′ revealed that compounds **7** and **8** have the same relative configuration. Therefore, on the basis of the positive optical rotation of **8** and biosynthetic considerations, the structure of **8** was deduced as (+)-(8*R*,7′*S*,8′*R*)-9′-*O*-(*E*)-cinnamoyl-5,5′-dimethoxylariciresinol.

Compound **9** was shown to have the molecular formula of C_33_H_38_O_12_, as established by the HRESIMS. The ^1^H- and ^13^C-NMR spectra of **9** closely resembled those of **7**, the only discernable difference being the presence of a new methoxy moiety and lack of a ketone moiety in **9**, suggesting that compound **9** contains a methoxy moiety rather than a ketone moiety at C-7. This was confirmed from the COSY correlation of H-7/H-8 and HMBC correlation of OMe/C-7. In the NOESY spectrum of **9**, the NOE correlations of H-7/H_2_-9′ and H-7′/H_2_-9′ also verified that H-7′ was oriented opposite to H-8 and H-8′. Thus, the structure of **9** was defined as 9′-*O*-(*E*)-feruloyl-5,7,5′-trimethoxy-lariciresinol.

The known compounds were identified as 1,2-dihydro-6,8-dimethoxy-7-hydroxy-1-(3,5-dimethoxy- 4-hydroxyphenyl)-*N*^1^,*N*^2^-bis-[2-(4-hydroxypeenyl)ethyl]-2,3-naphthalene dicarboxamide (**10**) [20], (+)-9,9′-*O*-di-(*E*)-feruloyl-5,5′-dimethoxy secoisolariciresinol (**11**) [25], (+)-9,9′-*O*-di-(*E*)-feruloyl-secoisolariciresinol (**12**) [12], (+)-9′-*O*-(*E*)-feruloyl-5,5′-dimethoxylariciresinol (**13**) [24], (+)-9′-*O*-(*E*)-feruloyl-5′-methoxylariciresinol (**14**) [26], (+)-5,5′-dimethoxylariciresinol (**15**) [27], (+)-5′-methoxylariciresinol (**16**) [28], arctigenin (**17**), matairesinol (**18**) [29], and (7*E*,8*R*′)-didehydroarctigenin (**19**) [30], respectively, by spectroscopic analysis and comparison of the data obtained with literature values.

### 2.2. Biological Activities of Compounds ***1***–***19***

#### 2.2.1. Cytotoxic Activity

The task of IC_50_ assessment for all isolates against human colon cancer (HCT-116), human non-small-cell lung carcinoma (NCI-H1650), and human ovarian cancer (A2780) cell lines began immediately following the purification and characterization of each lignan. 

Of the compounds, only 7′,9-epoxylignans with feruloyl or cinnamoyl group (compounds **7**–**9**, **13** and **14**) were selectively cytotoxic against NCI-H1650 cell line, with IC_50_ values of less than 20 μM. These results suggested the presence of a feruloyl or a cinnamoyl moiety at C-9′ in 7′,9-epoxylignans is essential for cytotoxicity against NCI-H1650 cell line. It is noteworthy that compound **7** displayed 4-6 folds more active than **8**, **9**, **13**, and **14**, indicating that the presence of the C-7 ketone could enhance the bioactivity. In addition, the dibenzylbutyrolactone lignans (**17**–**19**) exerted cytotoxicities against HCT-116 and A2780 cell lines, with IC_50_ values ranging from 0.28 to 18.47 μM (Table 3), but less potent than the positive control taxol (IC_50_ = 0.005 and 0.02 μM, respectively). Interestingly, the addition of the double bond at C-7−C-8 on **19** resulted in 4–40 folds less active than **17** and **18**. This implied that the C-7−C-8 double bond could reduce the cytotoxicity, especially against the A2780 cell line.

#### 2.2.2. Inhibitory Activity of Protein Tyrosine Phosphatase 1B

The isolates were also evaluated for inhibitory activities against protein tyrosine phosphatase 1B (PTP1B). Only compound **11** was moderate active toward PTP1B with an IC_50_ value of 13.5 μM. The positive control oleanolic acid gave an IC_50_ value of 3.82 μM.

#### 2.2.3. Anti-Inflammatory Activity

The inhibitory activity of compounds **1**–**19** against LPS-induced NO production in RAW264.7 macrophages was examined in this study. As a result, compounds **4**–**6**, **11** and **12** displayed inhibitions against LPS-induced NO production in RAW264.7 macrophages, with IC_50_ values of 46.8, 50.1, 58.6, 47.5, and 66.5 μM, respectively. Dexamethasone was used as positive control with an IC_50_ value of 9.5 μM.

## 3. Materials and Methods

### 3.1. General Experimental Procedures

Optical rotations were measured on an Autopol III automatic polarimeter (Rudolph Research, Hackettstown, NJ, USA). UV spectra were measured on a Cary 300 spectrometer (Agilent, Melbourne, Australia). ECD spectra were recorded on a J-815 spectrometer (JASCO, Tokyo, Japan). IR spectra were acquired on an Impact 400 FT-IR Spectrophotometer (Nicolet, Madison, WI, USA). Standard pulse sequences were used for all NMR experiments, which were run on either a Bruker spectrometer (600 MHz for ^1^H or 150 MHz for ^13^C, Karlsruhe, Germany) or a Varian INOVA spectrometer (500 MHz for ^1^H or 125 MHz for ^13^C, Palo Alto, CA, USA) equipped with an inverse detection probe. Residual solvent shifts for acetone-*d*_6_ were referenced to δ_H_ 2.05, δ_C_ 206.7 and 29.9, respectively. Accurate mass measurements were obtained on a Q-Trap LC/MS/MS (Turbo ionspray source) spectrometer (Sciex, Toronto, ON, Canada). Column chromatography (CC) was run using silica gel (200–300 mesh, Qingdao Marine Chemical Inc., Qingdao, China), and Sephadex LH-20 (Pharmacia Biotech AB, Uppsala, Sweden). HPLC separation was done on Waters HPLC components (Milford, MA, USA) comprising of a Waters 600 pump, a Waters 600 controller, a Waters 2487 dual λ absorbance, with GRACE preparative (250 × 19 mm) Rp C_18_ (5 μm) columns.

### 3.2. Plant Material

The twigs of *Litsea cubeba* were collected in Zhaotong, Yunnan Province, People’s Republic of China, in May 2013, and identified by Prof. Gan-Peng Li at Yunnan Minzu University. A herbarium specimen was deposited in at the Herbarium of the Department of Medicinal Plants, Institute of Materia Medica, Beijing 100050, People’s Republic of China (herbarium No. 2013-05-10).

### 3.3. Extraction and Isolation

The air-dried twigs of *L. cubeba* (12 kg) were ground and extracted using 30.0 L of 95% EtOH under ambient temperature for 3 × 48 h. The EtOH extract was concentrated in vacuo and the residue was suspended in H_2_O, then partitioned with EtOAc, to afford EtOAc and H_2_O soluble extracts.

The EtOAc fraction (300 g) was chromatographed over silica gel (1500 g), eluting with a gradient of acetone (0–100%) in petroleum ether, and 13 fractions (F_1_–F_13_) was obtained based on the TLC analysis. The F_9_ (12.0 g), which showed potent cytotoxicity against HCT-116, NCI-H1650, and A270 cell lines, was subjected to the reversed-phase flash chromatography over C-18 silica gel, eluting with a step gradient from 20 to 95% MeOH in H_2_O, to give 15 fractions (F_9-1_–F_9-15_). F_9-8_ (1.5 g) was separated on Sephadex LH-20 eluting with petroleum CHCl_3_-MeOH (1:1) to give three subfractions, and the first subfraction was purified by reversed-phase preparative HPLC (RP_18_, 5 μm, 254 nm, MeOH-H_2_O, 75:25) to yield **1** (9.2 mg). The second and third subfractions were further purified by preparative TLC developed with CHCl_3_-MeOH (15:1) to afford **15** (52 mg), **16** (35mg), and **18** (29 mg). F_9-9_ (1.0 g) was fractionated on a Sephadex LH-20 column using CHCl_3_-MeOH (1:1) as the eluent to yield five corresponding subfractions. Compound **10** (55 mg) was crystallized from a Me_2_CO solution of the second subfraction. The third subfraction was further purified by preparative TLC with CHCl_3_-MeOH (20:1) to give **17** (17 mg) and **19** (8 mg). The fourth subfraction was purified by reversed-phase preparative HPLC (RP_18_, 5 μm, 254 nm, MeOH-H_2_O, 85:15) to give **2** (56 mg), **3** (21 mg), and **14** (23 mg). Using the same HPLC system, the fifth subfraction afforded **7** (27 mg), **8** (12 mg) and **9** (8 mg), and **13** (17 mg). F_9-10_ (1.2 g) was chromatographed over Sephadex LH-20 eluting with CHCl_3_-MeOH (1:1), and then further separated by reversed-phase preparative HPLC (RP_18_, 5 μm, 254 nm, MeOH-H_2_O, 90:10), to afford **4** (8 mg) and **5** (5 mg). F_9-11_ (0.8 g) was fractionated on a Sephadex LH-20 column with CHCl_3_-MeOH (1:1) as the eluent to give three subfractions. The second and third subfractions were further purified by reversed-phase preparative HPLC (RP_18_, 5 μm, 254 nm, MeOH-H_2_O, 90:10) to afford **6** (12 mg), **11** (23 mg), and **12** (15 mg).

### 3.4. (−)-(7′R,8′S)-N-[2-(4-Hydroxyphenyl)-ethyl]-4,4′,9′-trihydroxy-3,5,3′,5′-tetramethoxy-2,7′-cyclo-lignan-7-en-9-amide *(**1**)*


White, amorphous powder.${\left[\alpha \right]}_{D}^{20}$ −35.0 (c 0.1, MeOH); UV (MeOH) λ_max_ (log ε) 204 (4.04), 200 (2.32), 245 2.12), 324 (1.13) nm; IR (KBr) ν_max_ 3372, 2935, 2849, 1643, 1611, 1516, 1459, 1427, 1329, 1286, 1218, 1115, 1030, 961, 912, 834, 646 cm^−1^; ^1^H-NMR (acetone-*d*_6_, 500 MHz) and ^13^C-NMR (acetone-*d*_6_, 125 MHz) data, see Table 1; Table 2; ESIMS *m*/*z* 574 [M + Na]^+^ and 550 [M − H]^−^; HRESIMS *m*/*z* 552.2234 [M + H]^+^ (calcd. for C_30_H_34_NO_9_, 552.2228) and 574.2048 [M + Na]^+^ (calcd. for C_30_H_33_NO_9_Na, 574.2048).

### 3.5. (−)-(7′R,8′S)-N^1^-[2-(4-Hydroxyphenyl)-ethyl]-N^2^-[2-(4-hydroxy-3-methoxyphenyl)-ethyl]-4,4′-dihydro-xy-3,5,3′,5′-tetramethoxy-2,7′-cyclolignan-7-en-9,9′-diamide *(**2**)*

White, amorphous power. ${\left[\alpha \right]}_{D}^{20}$ −23.0 (c 0.1, MeOH); UV (MeOH) λ_max_ (log ε) 204 (4.11), 250 (0.86), 281 (0.30), 328 (0.42) nm; IR (KBr) ν_max_ 3362, 2919, 2851, 1736, 1649, 1612, 1516, 1464, 1424, 1372, 1328, 1274, 1217, 1115, 1035, 890, 834, 802, 721, 640 cm^−1^; ^1^H-NMR (acetone-*d*_6_, 600 MHz) and ^13^C-NMR (acetone-*d*_6,_ 150 MHz) data, see Table 1; Table 2; ESIMS *m*/*z* 713 [M − H]^−^; HRESIMS *m*/*z* 713.2719 [M − H]^−^ (calcd. for C_39_H_41_N_2_O_11_, 713.2716).

### 3.6. (−)-(7′R,8′S)-N^1^-[2-(4-Hydroxy-3-methoxyphenyl)-ethyl]-N^2^-[2-(4-hydroxyphenyl)-ethyl]-4,4′-dihydro-xy-3,5,3′,5′-tetramethoxy-2,7′-cyclolignan-7-en-9,9′-diamide *(**3**)*


White, amorphous power. ${\left[\alpha \right]}_{D}^{20}$ −25.0 (c 0.1, MeOH); UV (MeOH) λ_max_ (log ε) 204 (4.12), 248 (0.82), 285 (0.27), 333 (0.45) nm; IR (KBr) ν_max_ 3391, 2920, 2851, 1647, 1611, 1541, 1517, 1465, 1425, 1367, 1278, 1203, 1116, 1035, 932, 888, 829, 801, 722, 650, 599 cm^−1^; ^1^H-NMR (acetone-*d*_6_, 600 MHz) and ^13^C-NMR (acetone-*d*_6_, 150 MHz) data, see Table 1; Table 2; ESIMS *m*/*z* ESIMS *m*/*z* 713 [M − H]^−^; HRESIMS *m*/*z* 713.2715 [M − H]^−^ (calcd. for C_39_H_41_N_2_O_11_, 713.2716).

### 3.7. (+)-(8S,8′S)-9-O-(E)-Cinnamoyl-secoisolariciresinol *(**4**)*

Yellow solid. ${\left[\alpha \right]}_{D}^{20}$ +18.2 (c 0.05, MeOH); UV (MeOH) λ_max_ (log ε) 204 (4.12), 230 (0.82), 287 (0.39), 329 (0.78) nm; IR (KBr) νmax 3391, 2920, 2850, 1683, 1645, 1608, 1516, 1463, 1428, 1375, 1341, 1272, 1237, 1155, 1119, 1033, 875, 820, 799, 721, 631 cm^−1^; ^1^H-NMR (acetone-*d*_6_, 500 MHz) and ^13^C-NMR (acetone-*d*_6_, 125 MHz) data, see Table 1; Table 2; ESIMS *m*/*z* 567 [M − H]^−^; HRESIMS *m*/*z* 569.2387 [M + H]^+^ (calcd. for C_31_H_37_NO_10_, 569.2381) and 591.2204 [M + Na]^+^ (calcd. for C_31_H_36_O_10_Na, 591.2201).

### 3.8. (+)-(8S,8′S)-9-O-(E)-Feruloyl-5,5′-dimethoxysecoisolariciresinol *(**5**)*

Yellow solid. ${\left[\alpha \right]}_{D}^{20}$ +22.2 (c 0.05, MeOH); UV (MeOH) λ_max_ (log ε) 206 (4.22), 234 (0.84), 284 (0.36), 326 (0.82) nm; IR (KBr) ν_max_ 3394, 2921, 2850, 1696, 1604, 1517, 1461, 1428, 1370, 1328, 1273, 1218, 1161, 1117, 1033, 984, 915, 825, 721, 645, 604 cm^−1^; ^1^H-NMR (acetone-*d*_6_, 600 MHz) and ^13^C-NMR (acetone-*d*_6_, 150 MHz) data, see Table 1; Table 2; HRESIMS *m*/*z* 621.2299 [M + Na]^+^ (calcd. for C_32_H_38_O_1__1_Na, 621.2306).

### 3.9. (+)-(8S,8′S)-9-O-(E)-Feruloyl-secoisolariciresinol *(**6**)*

Yellow solid. ${\left[\alpha \right]}_{D}^{20}$ +25.2 (c 0.1, MeOH); IR (KBr) ν_max_ 3367, 2928, 2855, 1683, 1601, 1516, 1454, 1431, 1375, 1271, 1207, 1154, 1033, 935, 846, 801, 724 cm^−1^; ^1^H-NMR (acetone-*d*_6_, 600 MHz) and ^13^C-NMR (acetone-*d*_6_, 150 MHz) data, see Table 1; Table 2; HRESIMS *m*/*z* 537.2134 [M − H]^−^ (calcd. for C_30_H_33_O_9_, 537.2130).

### 3.10. (+)-(8R,7′S,8′R)-9′-O-(E)-Feruloyl-5,5′-dimethoxylariciresinol-7-one *(**7**)*

Amorphous powder. ${\left[\alpha \right]}_{D}^{20}$ +19.5 (c 0.1, MeOH); UV (MeOH) λ_max_ (log ε) 211 (4.01), 234 (2.12), 318 (1.96) nm; ECD (MeOH) 331 (Δε − 0.37), 288 (Δε + 0.73), 222 (Δε + 2.01); IR (KBr) ν_max_ 3409, 2940, 2843, 1701, 1665, 1604, 1516, 1461, 1425, 1371, 1323, 1271, 1215, 1169, 1116, 1032, 983, 912, 845, 827, 765, 712, 662 cm^−1^; ^1^H-NMR (acetone-*d*_6_, 500 MHz) and ^13^C-NMR (acetone-*d*_6_, 125 MHz) data, see Table 1; Table 2; ESIMS *m*/*z* 609 [M − H]^−^; HRESIMS *m*/*z* 609.1980 [M − H]^−^ (calcd. for C_32_H_33_O_12_, 609.1978).

### 3.11. (+)-(8R,7′S,8′R)-9′-O-(E)-Cinnamoyl-5,5′-dimethoxylariciresinol *(**8**)*

Amorphous powder. ${\left[\alpha \right]}_{D}^{20}$ +23.0 (c 0.1, MeOH); IR (KBr) ν_max_ 3425, 2937, 2845, 1703, 1612, 1516, 1461, 1427, 1331, 1282, 1218, 1154, 1117, 1041, 980, 913, 832, 719 cm^−1^; ^1^H-NMR (acetone-*d*_6_, 600 MHz) and ^13^C-NMR (acetone-*d*_6_, 150 MHz) data, see Table 1; Table 2; HRESIMS *m*/*z* 625.2297 [M − H]^−^ (calcd. for C_33_H_37_O_12_, 625.2291).

### 3.12. 9′-O-(E)-Feruloyl-5,7,5′-trimethoxylariciresinol *(**9**)*

Amorphous powder. ${\left[\alpha \right]}_{D}^{20}$ +21.0 (c 0.1, MeOH); IR (KBr) ν_max_ 3395, 2933, 2849, 1701, 1610, 1517, 1462, 1428, 1372, 1324, 1270, 1214, 1159, 1116, 1033, 983, 909, 831, 703 cm^−1^; ^1^H-NMR (acetone-*d*_6_, 500 MHz) and ^13^C-NMR (acetone-*d*_6_, 125 MHz) data, Table 1; Table 2; HRESIMS *m*/*z* 625.2297 [M − H]^−^ (calcd. for C_33_H_37_O_12_, 625.2291).

### 3.13. Cytotoxicity Assay

The cytotoxic activity was determined against human colon cancer (HCT-116), human non-small-cell lung carcinoma (NCI-H1650), and human ovarian cancer (A2780) cell lines which were bought from the Cell Bank of Shanghai Institute of Cell Biology (Chinese Academy of Sciences) and originally obtained from the American Type Culture Collection (ATCC, Rockville, MD, USA). Cells were grown in RPMI 1640 (GIBCO, New York, NY, USA) supplemented with 10% fetal calf serum (Life Technologies, Carlsbad, CA, USA), penicillin G (100 U/mL), and streptomycin (100 μg/mL) at 37 °C in a 5% CO_2_ and seeded in 96-well plates (CLS3635, Corning^®^, Sigma, Santa Clara, CA, USA) at a cell density of 3000 per well over night, and then were treated with various diluted concentrations (each concentration was arranged triple) of compounds **1**–**19**, which were prepared with DMSO (Sigma) to 100 μM stock solution and stored in −20 °C in advance. After 24 h of treatment, 10 μL of MTT (5 mg/mL in PBS) was then added directly to all wells and the plates were placed in the dark at 37 °C for 3 h incubation. Cell viability was measured by observing absorbance at 570 nm on a SpectraMax^190^ microplate reader (Molecular Devices, Silicon Valley, CA, USA). IC_50_ values were calculated using Microsoft Excel software (version 2010, Redmond, WA, USA). Taxol was used as a positive control.

### 3.14. PTP1B Inhibition Assay

The recombinant GST-hPTP1B (gluthathione *S*-transferase-human protein tyrosine phosphatase 1B) bacteria pellets were purified by a GST bead column. The dephosphorylation of *para*-nitrophenyl phosphate (*p*-NPP) was catalyzed to *para*-nitrophenol by PTP1B. Enzyme activity involving an end-point assay, which intensified the yellow color, was measured at a wavelength of 405 nm. All compounds were dissolved in 100% dimethyl sulfoxide (DMSO), and reactions, including controls, were performed at a final concentration of 10% DMSO. Selected compounds were first evaluated for their ability to inhibit the PTPase reaction at a 10 μM concentration at 30 °C for 10 min, in a reaction system with 3 mM *p*-NPP in HEPES assay buffer (pH 7.0). The reaction was initiated by addition of the enzyme and quenched by addition of 1 M NaOH. The amount of the produced *p*-nitrophenol was determined at 405 nm using a microplate spectrophotometer (uQuant, Bio-Tek, Winooski, VT, USA). IC_50_ values were evaluated using a sigmoidal dose-response (variable slope) curve-fitting program of GraphPad Prism 4.0 software (La Jolla, CA, USA). Oleanolic acid was used as a positive control.

### 3.15. Nitric Oxide (NO) Production in RAW264.7 Macrophages

The RAW 264.7 macrophages were cultured in The RPMI 1640 medium (Hyclone, Logan, UT, USA) containing 10% FBS. The compounds were dissolved in DMSO and further diluted in medium to produce different concentrations. The cell mixture and culture medium were dispensed into 96-well plates (2 × 105 cells/well) and maintained at 37 °C under 5% CO2. After preincubation for 24 h, serial dilutions of the test compounds were added into the cells, up to the maximum concentration 25 μM, then added with LPS to a concentration 1 μg/mL and continued to incubate for 18 h. The amount of NO was assessed by determined the nitrite concentration in the cultured RAW264.7 macrophage supernatants with Griess reagent. Aliqueots of supernatants (100 μL) were incubated, in sequence, with 50 μL 1% sulphanilamide and 50 μL 1% naphthylethylenediamine in 2.5% phosphoric acid solution. The sample absorbance was measured at 570 nm by a 2104 Envision Multilabel Plate Reader (PerkinElmer, Inc., Waltham, MA, USA). Dexamethasone was used as a positive control.

## 4. Conclusions

In summary, bioassay-guided isolation of cytotoxic fractions of the twigs of *L. cubeba* revealed the presence of nine new lignans **1**–**9** and ten analogues **10**–**19**. Initially, all of the isolated compounds were evaluated against HCT-116, NCI-H1650, and A2780 tumor cell lines. Of the compounds, only 7′,9-epoxylignans with feruloyl or cinnamoyl group (**7**–**9**, **13** and **14**) were selectively cytotoxic against NCI-H1650 cell line, with IC_50_ values of less than 20 μM, whereas, the dibenzylbutyrolactone lignans **17**–**19** exerted cytotoxicity against HCT-116 and A2780 cell lines, with IC_50_ values ranging from 0.28 to 18.47 μM. The results highlighted the structure-activity relationship importance of a feruloyl or a cinnamoyl moiety at C-9′ or/and C-7 ketone in 7′,9-epoxylignans. The isolates were also examined for inhibitory activities against PTP1B and LPS-induced NO production in RAW264.7 macrophages. As a result, compound **11** was moderate active toward PTP1B with an IC_50_ value of 13.5 μM and compounds **4**–**6**, **11** and **12** displayed inhibitions against LPS-induced NO production in RAW264.7 macrophages, with IC_50_ values of 46.8, 50.1, 58.6, 47.5, and 66.5 μM, respectively. The present results provide additional phytochemical and bioactive information of this medicinal and spiced plant.

## Figures and Tables

**Figure 1 molecules-24-00306-f001:**
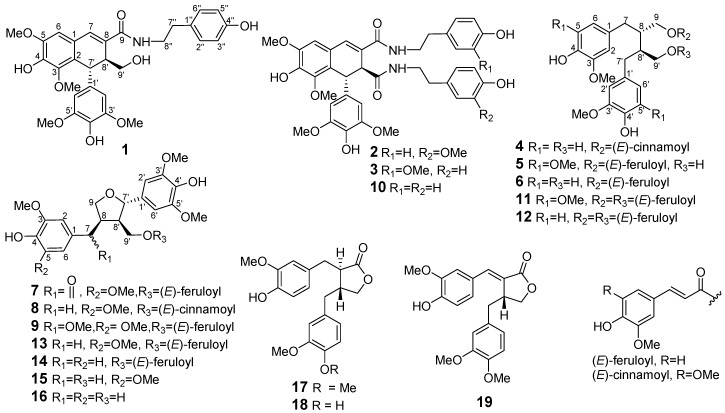
The structures of compounds **1**–**19.**

**Figure 2 molecules-24-00306-f002:**
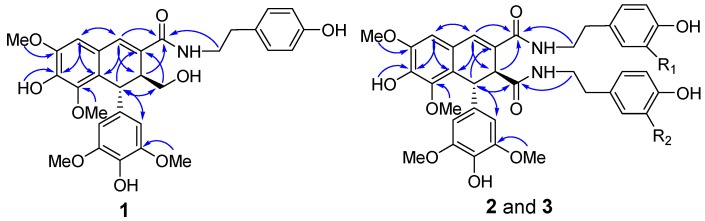
The key HMBC correlations of **1**–**3**.

**Figure 3 molecules-24-00306-f003:**
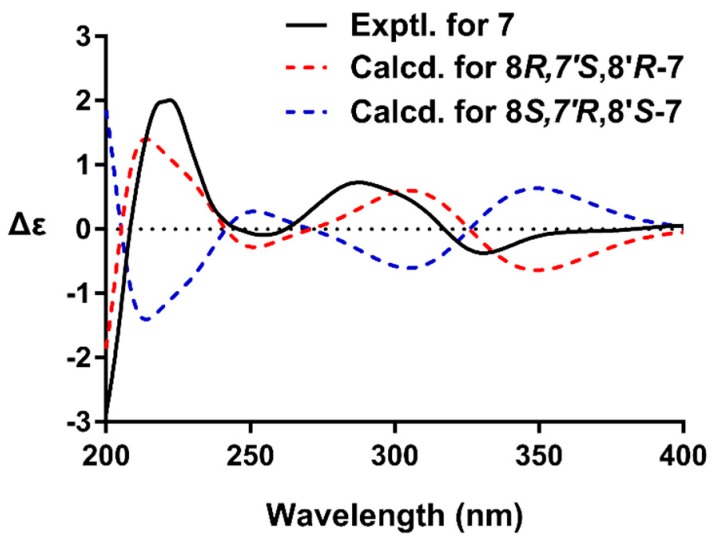
The experimental ECD spectrum of **7** (black), and the calculated ECD spectra of (8*R*,7′*S*,8′*R*)-**7** (red) and (8*S*,7′*R*,8′*S*)-**7** (blue).

**Table 1 molecules-24-00306-t001:** ^1^H-NMR Data (δ_H_ (mult, *J*, Hz)) of Compounds **1**–**9** in Acetone-*d*_6_
*^a^*.

No.	1	2	3	4	5	6	7	8	9
2				6.71 d (1.5)	6.42 s	6.70 d (1.8)	7.39 s	6.57 s	6.67 s
5				6.71 d (7.5)		6.71 d (7.8)			
6	6.74 s	6.69 s	6.60 s	6.61 dd (7.5, 1.5)	6.42 s	6.61 dd (7.8, 1.8)	7.39 s	6.57 s	6.67 s
7	7.19 s	7.18 s	7.21 s	2.80 dd (13.5, 7.0); 2.62 dd (13.5, 8.0)	2.79 dd (14.2, 7.2); 2.62 dd (14.2, 8.4)	2.80 dd (13.8, 6.6); 2.62 dd (13.8, 8.4)		2.91 dd (13.2, 5.4); 2.59 dd (13.2, 10.2)	4.35 d (6.5)
8				2.32 m	2.31 m	2.31 m	4.57 m	2.82 m	2.84 m
9				4.36 dd (11.5, 6.5); 4.11 dd (11.5, 6.0)	4.42 dd (10.8, 6.0); 4.10 dd (10.8, 6.0)	4.36 dd (11.4, 6.6); 4.11 dd (11.4, 6.0)	4.35 t (8.0); 4.22 t (8.0)	4.04 dd (8.4, 6.6); 3.74 dd (8.4, 6.6)	4.14 t (8.5); 4.04 t (8.5)
2′	6.39 s	6.38 s	6.38 s	6.73 d (1.5)	6.44 s	6.73 d (1.8)	6.78 s	6.68 s	6.63 s
5′				6.69 d (7.5)		6.69 d (7.8)			
6′	6.39 s	6.38 s	6.38 s	6.61 dd (7.5, 1.5)	6.44 s	6.61 dd (7.8, 1.8)	6.78 s	6.68 s	6.63 s
7′	4.62 s	5.03 s	5.03 s	2.70 dd (13.5, 7.0); 2.63 dd (13.5, 8.0)	2.70 dd (14.2, 7.2); 2.63 dd (14.2, 8.4)	2.70 dd (13.8, 6.6); 2.63 dd (13.8, 8.4)	4.74 d (7.5)		4.82 d (5.5)
8′	3.14 dd (7.5, 7.5)	3.66 s	3.67 s	1.99 m	1.99 m	1.99 m	3.01 m	2.61 m	
9′	3.59 m 3.28 m			3.67 m; 3.59 m	3.67 m; 3.61 m	3.69 m; 3.59 m	4.16 d (6.5)	4.53 dd (11.4, 6.6); 4.30 dd (11.4, 7.8)	
2′′	6.98 d (8.5)	6.98 d (8.5)	6.79 d (1.8)	7.00 s	7.32 d (1.8)	7.32 d (1.8)	7.06 d (1.5)	6.98 s	7.27 d (2.0)
3′′	6.71 d (8.5)	6.72 d (8.5)							
5′′	6.71 d (8.5)	6.72 d (8.5)	6.71 d (7.8)		6.86 d (8.4)	6.81 d (8.4)	6.82 d (8.5)		6.85 d (8.0)
6′′	6.98 d (8.5)	6.98 d (8.5)	6.61 dd (7.8, 1.8)	7.00 s	7.13 dd (8.4, 1.8)	7.13 dd (8.4, 1.8)	6.96 dd (8.5, 1.5)	6.98 s	7.11 dd (8.0, 2.0)
7′′	2.69 t (7.5)	2.70 t (7.0)	2.72 t (7.2)		7.58 d (15.6)	7.57 d (15.6)	7.16 d (16.0)	7.47 d (16.2)	7.49 d (15.5)
8′′	3.39 dt (7.4, 4.5)	3.41 t (6.0)	3.47 m, 3.39 m		6.42 d (15.6)	6.41 d (15.6)	5.89 d (16.0)	6.39 d (16.2)	6.34 d (15.5)
2′′′		6.79 d (1.5)	6.93 d (8.4)						
3′′′			6.70 d (8.4)						
5′′′		6.69 d (8.0)	6.70 d (8.4)						
6′′′		6.55 dd (8.0, 1.5)	6.93 d (8.4)						
7′′′		2.58 t (7.0)	2.56 t (7.2)						
8′′′		3.28 t (7.0)	3.29 m, 3.21 m						
OMe-3	3.58 s	3.69 s	3.69 s	3.75 s	3.73 s	3.75 s	3.84 s	3.79 s	3.80 s
OMe-5	3.86 s	3.85 s	3.85 s		3.73 s		3.84 s	3.79 s	3.80 s
OMe-7									3.17 s
OMe-3′	3.67 s	3.67 s	3,67 s	3.75 s	3.73 s	3.75 s	3.83 s	3.88 s	3.77 s
OMe-5′	3.67 s	3.67 s	3.67 s		3.73 s		3.83 s	3.88 s	3.77 s
OMe-3′′			3.78 s	3.88 s	3.91 s	3.90 s	3.91 s	3.79 s	3.91 s
OMe-5′′				3.88 s				3.79 s	
OMe-3′′′		3.80 s							
OH-4	7.76 s	7.78 s	7.79 s	7.29 s	6.91 s	7.27 s		7.09 s	
OH-4′	6.90 s	6.91 s	6.91 s	7.26 s	6.89 s	7.24 s		6.98 s	
OH-4′′		8.08 s	7.21 s	7.75 s	8.15 s	8.12 s		7.77 s	
OH-4′′′		7.26 s	8.07 s						
NH	7.45 t (4.5)	7.81 t (4.5), 7.59 t (4.5)	7.72 t (4.5), 7.59 t (4.5)						

*^a^*^1^H-NMR data (δ) were measured at 600 MHz or 500 MHz. The assignments were based on ^1^H-^1^H COSY, HSQC, and HMBC experiments.

**Table 2 molecules-24-00306-t002:** ^13^C-NMR Data (δ_C_) for Compounds **1**–**9** in Acetone-*d*_6_
*^a^*.

No.	1	2	3	4	5	6	7	8	9
1	132.0	123.8	123.8	132.9	132.4	132.9	129.6	131.8	134.5
2	124.6	126.5	126.5	113.2	107.2	113.2	107.2	106.9	106.9
3	147.0	146.4	146.4	148.1	148.5	148.1	148.4	148.9	148.5
4	141.8	142.4	142.4	145.5	134.9	145.5	142.2	135.2	136.0
5	148.2	148.1	148.1	115.5	148.5	115.5	148.4	148.9	148.5
6	108.0	108.3	108.2	122.3	107.2	122.3	107.2	106.9	106.9
7	131.5	132.5	133.6	35.4	35.9	35.4	198.2	34.2	82.6
8	124.4	128.3	128.4	40.7	40.6	40.8	47.6	43.6	48.1
9	169.1	169.8	169.6	65.2	65.2	65.2	71.1	73.3	70.3
1′	135.9	135.1	135.1	133.4	131.8	133.4	132.9	134.6	131.7
2′	106.4	106.4	106.4	113.2	107.3	113.2	104.7	104.3	104.4
3′	148.3	148.4	148.4	148.1	148.5	148.1	148.6	148.7	148.8
4′	135.3	135.5	135.5	145.6	135.0	145.5	136.3	136.0	136.5
5′	148.3	148.4	148.4	115.4	148.5	115.4	148.6	148.7	148.8
6′	106.4	106.4	106.4	122.3	107.3	122.3	104.7	104.3	104.4
7′	39.0	39.5	39.6	34.9	35.4	35.0	84.9	84.5	85.1
8′	46.1	49.1	49.1	44.1	44.1	44.2	51.5	50.3	49.4
9′	64.6	171.4	171.4	62.1	62.1	62.1	62.8	63.4	63.6
1′′	131.2	131.1	131.7	126.1	127.4	127.5	127.2	126.0	127.3
2′′	130.5	130.6	113.1	106.8	111.3	11.3	111.0	106.7	111.3
3′′	116.0	116.1	148.2	148.9	148.7	148.8	148.6	148.6	148.7
4′′	156.6	156.7	145.9	139.4	150.1	150.1	149.9	139.5	150.1
5′′	116.0	116.1	115.7	148.9	116.1	116.0	115.9	148.6	116.1
6′′	130.5	130.6	122.0	106.8	123.9	124.0	123.7	106.7	123.8
7′′	35.6	35.5	36.0	145.9	145.6	145.6	145.6	146.2	145.8
8′′	42.2	42.4	42.3	116.2	116.0	116.0	115.1	115.9	114.8
9′′				167.5	167.6	167.5	166.7	167.3	167.3
1′′′		131.8	131.2						
2′′′		113.0	130.6						
3′′′		148.2	116.0						
4′′′		145.8	156.6						
5′′′		115.6	116.0						
6′′′		122.0	130.6						
7′′′		36.1	35.7						
8′′′		41.9	42.1						
OMe-3	60.4	60.3	60.3	56.5	56.5	56.1	56.7	56.6	56.6
OMe-5	56.5	56.2	56.2		56.5		56.7	56.6	56.6
OMe-7									56.1
OMe-3′	56.6	56.7	56.7	56.1	56.4	56.1	56.6	56.7	56.6
OMe-5′	56.6	56.7	56.7		56.4		56.6	56.7	56.6
OMe-3′′			56.5	56.7	56.3	56.3	56.3	56.6	
OMe-5′′				56.7				56.6	
OMe-3′′′		56.6							

*^a^*^13^C-NMR data (δ) were measured at 150 MHz or 125 MHz. The assignments were based on ^1^H-^1^H COSY, HSQC, and HMBC experiments.

**Table 3 molecules-24-00306-t003:** Cytotoxicity of Compounds **1**–**19** to HCT-116, NCI-H1650, and A2780 Cell Lines.

Compound	IC_50_ (μM)		
	HCT-116	NCI-H1650	A2780
**1**	>20	>20	>20
**2**	>20	>20	>20
**3**	>20	>20	>20
**4**	>20	>20	>20
**5**	>20	>20	>20
**6**	>20	>20	>20
**7**	>20	2.47	>20
**8**	>20	11.25	>20
**9**	>20	13.16	>20
**10**	>20	>20	>20
**11**	>20	>20	>20
**12**	>20	>20	>20
**13**	>20	9.68	>20
**14**	>20	10.52	>20
**15**	>20	>20	>20
**16**	>20	>20	>20
**17**	3.25	>20	0.28
**18**	13.95	>20	1.53
**19**	18.47	>20	12.8
Taxol ^a^	0.005	1.28	0.02

^a^ Taxol was used as a positive control.

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
