# Peer review of "Lignans from the Twigs of Litsea cubeba and Their Bioactivities"

_molecules, 2019, doi:10.3390/molecules24020306_

Round 1

Reviewer 1 Report

Revision of the manuscript “molecules-419849”

General comments

The manuscript “Lignans from the Twigs of Litsea cubeba and Their Bioactivities” by Li et al. describes the isolation and structural elucidation of 9 phenolic analogs along with 10 known compounds. Different spectroscopic methods were used, including NMR spectroscopy, mass spectrometry, and IR spectrophotometry. Furthermore, absolute configurations were established based on optical rotation comparison with known analogs, except for 7, were theoretical (DFT) and experimental ECD spectra were compared. All the isolated compounds were studied using an array of biological assays, but only a few compounds were shown to be potent. This manuscript will be suitable for publication in Molecules, after the authors consider the following comments. Additionally, the authors are strongly encouraged to do extensive editing of English language.

Specific Comments

 Line 59: Since F in F9 stands for “fraction”, the sentence “… fraction F9 …” is repetitive. To be checked elsewhere in the manuscript

Line 107: There are multiple instances where the compound numbers seem incorrect. Here, I guess the authors have found 2 being very similar to 10, not 9. It should be verified everywhere.

Line 109: The listing of all the NMR spectroscopic data in the text is not useful here.

Line 111: The HMBC correlation from H2-7’’’ and NH to C-9’ is difficult to interpret since the numbering of the N-[2-(4-hydroxyphenyl)-ethyl] groups is not explicated. Also, the NH should be assigned since they show different chemical shifts.

Line 162: The sentence “These data demonstrated the presence of C-7 ketone in 7.” is confusing since there is no data given before the sentence.

Line 163: The coupling constant is for H-7’, not H-7.

Line 170: 8R’ should be replaced by 8’R, everywhere in the text

Line 178: A NOESY correlation between H2-9/H2-9’ do not support any configuration.

Line 182: I assume NMR spectra of 9 closely resembled those of 7, not 6.

Line 183: The authors are encouraged to establish the configuration at C-7.

Line 197: The header for NCI is incomplete. Furthermore, pharmacological data for PTP1B inhibition and anti-inflammatory activity are missing.

Line 239: The complete taxonomic identification (no abbreviation, with authority) should be given.

Author Response

Responses

Reviewer 2 Report

The authors report studies investigating bioactivities of lignans from twigs of Listea cubeba.  The results are interesting and generally well supported.  Additional revisions would improve the manuscript.

Specific Comments

In Section 2.2.3, the authors report anti-inflammatory activity, which  should be shown in separate Table 4, because Table 3 includes only data on cytotoxicity. Please, add Table 4 with data on anti-inflammatory activity.

The authors study toxicity against several cell lines and then antiinflammatory activity in RAW cells.  These is a mouse macrophage cell line. What species are the tumor cell lines from?  This should be clarified.  It would be more informative to use a human macrophage cell line (for example THP-1) for comparison instead of crossing species for the various tumor cells and macrophages.

There are a number of grammatical errors that need to be corrected.

Author Response

Here are Responses to the Reviewer 2 Comments

1. In Section 2.2.3, the authors report anti-inflammatory activity, which should be shown in separate Table 4, because Table 3 includes only data on cytotoxicity. Please, add Table 4 with data on anti-inflammatory activity.

Thank you for your suggestion. However, the anti-inflammatory activity data, including the positive control data, have been clearly shown in the text (please see lines 214-219. In order to limit the length of the manuscript, we did not add table 4 with the anti-inflammatory activity data.

2. The authors study toxicity against several cell lines and then antiinflammatory activity in RAW cells.  These is a mouse macrophage cell line. What species are the tumor cell lines from?  This should be clarified.  It would be more informative to use a human macrophage cell line (for example THP-1) for comparison instead of crossing species for the various tumor cells and macrophages.

Thank you for your so many valuable suggestions. The tumor cell lines in the cytotoxicity assay were obtained from human cancer cell lines. These have been clearly shown the text and experimental section.

Yes! You are right. A human macrophage cell line such as THP-1 would be more informative to demonstrate the activity. However, this is only a preliminary screening. If we found a promising active compound, we should use a human macrophage cell line for comparison instead of crossing species for the various tumor cells and macrophages

Round 2

Reviewer 2 Report

The authors addressed my concerns.